# Barriers and facilitators for adopting a healthy lifestyle in a multi-ethnic population: A qualitative study

**Mythily Subramaniam** [1,2]*, **Fiona Devi**[1], **P. V. AshaRani**[1], **Yunjue Zhang**[1], **Peizhi Wang**[1], **Anitha Jeyagurunathan**[1], **Kumarasan Roystonn** [1], **Janhavi Ajit Vaingankar**[1], **Siow Ann Chong**[1]

**1** Research Division, Institute of Mental Health, Singapore, Singapore, **2** Saw Swee Hock School of Public Health, National University of Singapore, Singapore, Singapore

* Mythily@imh.com.sg

**Data Availability Statement:** Data cannot be shared publicly because of ethical and institutional regulations. Data are available from the Institute of Mental Health Institutional Research Review

## Abstract

A healthy lifestyle is defined as 'a way of living that lowers the risk of being seriously ill or dying early.' Although barriers and facilitators of healthy lifestyles have been well-studied among populations like those with chronic non-communicable diseases, adolescents, and older adults in Asia, less information is available on barriers and facilitators perceived by the general adult population. Using a qualitative methodology and leveraging the socio-ecological model, the current study aimed to understand the barriers and facilitators of a healthy lifestyle in a sample of Singapore residents. Overall, 30 semi-structured interviews were conducted in English and other local languages from August 2020 to March 2021. Transcripts were analysed using framework analysis. Five main themes pertaining to personal, interpersonal, environmental, socio-cultural, and policy-level factors were classified under the two overarching categories of barriers and facilitators of healthy lifestyles. The results of this study offer important insights into understanding the barriers and facilitators to the adoption of a healthy lifestyle among people in Singapore. Furthermore, our findings illustrate the complex interplay between individuals, social relationships, environment, and policy that can act as either a barrier or a facilitator to adopting a healthy lifestyle.

## Introduction

Life expectancy at birth has increased globally, and improvements in population health status have been observed for every life stage [1]. Public health initiatives such as universal immunisation, enhanced health infrastructure, improved maternal and infant health, tobacco control, and motor vehicle safety have driven these changes globally [2]. This increased life expectancy is associated with changing disease patterns, i.e., a transition from acute to chronic diseases as the primary source of morbidity and mortality worldwide [3]. At the same time, the physician-patient relationship has progressively moved towards shared decision-making, where clinicians and patients make decisions together using the best available evidence [4]. Furthermore, with patients becoming avid consumers of medical knowledge and taking on a more active role in their well-being, the responsibility for health rests on individuals and societies [5].

Committee (contact via research@imh.com.sg) for researchers who meet the criteria for access to confidential data.

**Funding:** SAC received the funding NMRC/HSRG/0085/2018 This study is supported by the Singapore Ministry of Health's National Medical Research Council under its Health Services Research Grant https://www.nmrc.gov.sg/who-we-are The funders had no role in study design, data collection and analysis, decision to publish, or preparation of the manuscript.

**Competing interests:** The authors have declared that no competing interests exist.

The World Health Organisation (WHO) defines a healthy lifestyle as 'a way of living that lowers the risk of being seriously ill or dying early' [6]. Pender's health promotion model defines barriers to a healthy lifestyle as 'factors that directly interfere with the enactment of a health-promoting behavior or mediate by reducing the commitment to the plan of action for changing behavior' [7]. On the other hand, facilitators are defined as factors that promote or enable the uptake and maintenance of a healthy lifestyle [8]. Identifying the barriers and facilitators can guide the design and implementation of evidence-informed behaviour change interventions that can specifically leverage the facilitators and target the barriers across multiple levels to improve healthy lifestyles.

A systematic review of 32 studies from 1980 to February 2010 by Murray et al. [9] found that a better understanding of illness, and perception of significant consequences of illness, were strong facilitators that promoted the uptake of lifestyle behavior change interventions. While stress, depression, lack of social support, and transport and commute time problems were significant barriers. Similar findings were reported by Kelly et al. [10] in their rapid systematic review, which examined barriers and facilitators to the uptake and maintenance of healthy behaviours by people in their mid-life. Across 28 qualitative studies, 11 longitudinal cohort studies, and 46 systematic reviews, the authors identified several barriers. These included lack of time, access (to transport, facilities, and resources), financial costs, entrenched attitudes and behaviours, low socioeconomic status, and lack of knowledge. In contrast, facilitators included enjoyment and a sense of well-being associated with physical activity, health benefits including healthy ageing, social support, clear messages, accessible websites, and previous experience of ill health.

In Asia, barriers and facilitators of healthy lifestyles have been well-studied among populations like those with chronic non-communicable diseases, adolescents, and older adults. However, less information on barriers and facilitators perceived by the general adult population is available. A qualitative study from Sri Lanka identified several barriers to adopting physical activity in an urban activity-friendly area. The barriers included competing priorities like work, physical concerns like safety, health concerns such as discomfort, resources including facilities and social support, and lack of understanding of the importance of physical activity for health [11]. A study from Nepal that examined the barriers and facilitators to healthy eating in a worksite cafeteria identified the unavailability or high cost of healthy foods, cultural preferences for fried food, and difficulty in changing eating habits as the main barriers. This study identified the availability and affordability of healthy food as the main facilitators of healthy eating [12]. A qualitative study conducted in India among the attendees of an urban health centre identified motivation/willpower, time management skills, knowledge and perceived benefits of physical activity, health problems, and availability of exercise facilities as facilitators of physical activity. Lack of time, space, or equipment, unfavourable weather, physical restriction, and laziness were important barriers [13].

Singapore is a densely populated, urbanised, city-state in Southeast Asia with a multiracial population of about 5.5 million, comprising Chinese, Malays, Indians, and a smaller proportion of other ethnicities [14]. In the past three decades, life expectancy in Singapore has risen by about ten years; however, healthy life expectancy at birth increased only by 7.2 years [15]. Thus, there is a need to promote successful ageing in the population [16].

The Singapore Government has made concerted efforts and worked consistently across sectors to promote evidence-based practices to build and maintain a culture of active living in the population [17]. In addition, Singapore supports the philosophy of individual responsibility, and it *remains a central tenet of Singapore's approach to healthcare* [18]. In 2016, the Singapore Health Minister declared War on Diabetes (WoD). The effort was mainly in response to the higher prevalence of diabetes in Singapore compared to the global prevalence rate, with nearly

one in ten Singaporeans (9.5%) suffering from the disease [19]. As part of this initiative, the government increased the availability and accessibility of physical activity programmes, launched nationwide physical activity-based challenges, and increased the availability of healthier food options in schools, restaurants, and food courts [20]. Thus, against this backdrop of sustained health-promoting policies and the focus on personal responsibility for health, Singapore presents a unique opportunity to understand the factors related to the uptake of a healthy lifestyle. However, no study has examined the barriers and facilitators for adopting a healthy lifestyle in Singapore to date.

Several theoretical frameworks have been offered to explain health behavior. The socio-ecological model (SEM) has been widely used to understand the interrelations between personal, social, and environmental determinants of lifestyle behavior [21, 22]. The model is attractive as it incorporates, intrapersonal or personal (biological, psychological), interpersonal/cultural, organizational, physical environment (built, natural), and policy (laws, regulations) influences. We adopted the perspective of the SEM framework in our qualitative inquiry to gain a deeper understanding of the barriers and facilitators of a healthy lifestyle in Singapore. Since ecological models incorporate a wide range of influences at multiple levels and explicitly include environmental and policy variables that are expected to influence behavior, the researchers felt that it would be most appropriate for the current inquiry. A secondary aim of the study was to explore if the people were aware of and utilized the interventions launched as part of the WoD to improve healthy lifestyles. The findings of such a study can help develop targeted interventions to overcome the barriers and enhance the facilitators to improve the impact of national campaigns in Singapore.

## Methods

### Study design and setting

The data for the current study was part of a more extensive study that examined the knowledge, attitudes, and protective practices toward diabetes among the public in Singapore. The study comprised a quantitative survey (n = 2895) and a qualitative phase (n = 30) to explore the barriers and facilitators of a healthy lifestyle in Singapore. The study methodology was published in an earlier article [23]. The sample for the nationwide survey was derived using a disproportionate stratified sampling design. In all, 12 strata: a combination of 3 strata for ethnicities (Chinese, Malay, and Indian) and four strata for age (18 to 34 years, 35 to 49 years, 50 to 64 years, and 65 years and above) were employed for the sampling. The proportion of respondents in each ethnic group (Chinese, Malay, and Indian) was set at approximately 30%, while the proportion of respondents in each age group was set at around 20% to ensure a sufficient sample size for these population subgroups [23].

The participants for the qualitative study were recruited from among those who participated in the quantitative survey and permitted recontact for future research studies. The inclusion criteria comprised Singapore citizens or permanent residents aged 21 years or above, the ability to speak in either English, Chinese, Malay, or Tamil, and not being diagnosed by a doctor as having diabetes. Participants were stratified according to their age ($\geq$ 40 years and $<$ 40 years), gender, and ethnicity (Chinese, Malay, Indian, and others), and then randomly chosen using an online randomisation software and recruited into the qualitative phase. To account for participant refusals, the sample was drawn in excess (i.e., 60 English-speaking participants and 30 native-language speakers). This sampling allows multiple perspectives to be presented that illustrate the complexity of the phenomenon under study [24]. However, the researchers did not link the quantitative data provided by the individuals with the qualitative data.

## Data collection

In Singapore, The Government imposed a nationwide 'circuit breaker' comprising restrictions on public gatherings and dining in restaurants, shift to home-based schooling, and working from home from 7th April 2020 until 1st June 2020. As of 2nd June 2020, Singapore entered the 'reopening' phase, and businesses and activities were progressively allowed to operate. The study period for the current study was eight months, from August 2020 to March 2021. While the measures had been relaxed, and a significant proportion of the population was vaccinated, some participants were uncomfortable doing face-to-face interviews. Interviews were therefore conducted in person or via the Zoom video-conferencing platform, depending on the participants' preference.

In all, 30 interviews were conducted; 20 were in English, four in Chinese, and three in Malay and Tamil. Each interview lasted between 60–90 minutes and was audio-recorded. Participants were interviewed in venues that afforded privacy so that they could freely express their views. Participants who opted for a Zoom interview were similarly informed that they should ensure a quiet and private interview setting. Using an interview guide, a trained qualitative interviewer, accompanied by a note-taker, conducted the interviews.

The interview guide was developed based on existing literature [25, 26] and further modified after discussions with a general practitioner and a diabetologist. The researchers minimally modified the interview guide to include relevant probes after the first two interviews (S1 File). A guided discussion format was used, and participants were encouraged to speak freely about their thoughts and experiences [27].

Data analysis was undertaken concurrently, allowing emerging themes to inform ongoing data collection. The researchers met up regularly and discussed emerging findings to ensure the trustworthiness of the data. Data collection ceased when saturation could reasonably be assumed. The team members transcribed and analysed the data after recruiting the first 20 English-language interviews before commencing the local-language interviews. This was to ensure that we had reached thematic saturation with the data collection and to simultaneously assess the language-specific interviews for the emergence of any new themes. An additional ten interviews were conducted with native speakers, i.e., those able to speak only in Chinese, Malay, and Tamil. This ensured that the perspectives of those belonging to a different socio-cultural background were taken into consideration.

Written informed consent was taken from all the participants, and ethical approval for the study was obtained from the relevant institutional review board (National Healthcare Group Domain Specific Review Board; protocol ref:2019/00926).

## Qualitative analysis

The interviews (English and language-specific) were transcribed or translated and transcribed verbatim by a professional transcription firm and checked for accuracy by a study team member. Transcripts were analysed using framework analysis. The framework method was initially developed for large-scale policy research [28]; however, it is now widely used in healthcare research. It is a data analysis method rather than a research paradigm which, unlike entirely inductive and iterative approaches, may be shaped by existing ideas and is less focused on producing a new theory [29].

## Data familiarisation

First, seven researchers (ZYJ, AR, FD, WP, KR, AJ, and MS) familiarised themselves with the first eight transcripts by reading them multiple times. This was in line with Srivastava and Thomson [30], who stated that, given the large volume of data in qualitative research, not

every piece of material may be reviewed at this stage. Following the deep reading, initial themes were identified by individual researchers. Next, these themes were checked against the interview guide and study objectives, resulting in the development of a set of preliminary codes for different barriers and facilitators to a healthy lifestyle.

## Constructing an initial thematic framework

The researchers then met to discuss and combine their preliminary codes. These discussions helped in resolving disagreements in defining or including themes. Largely there was consensus among the team members, but when there was a disagreement that could not be resolved, the first author made a call on the inclusion and definition of initial themes. These initial codes were then sorted into a hierarchy of themes and sub-themes to construct an initial framework. To ensure that all the research objectives were met, the initial framework consisted of four main categories, with several sub-themes under each category.

On reaching a consensus, a codebook was constructed, which contained a detailed description of each code, the inclusion and exclusion criteria, and typical and atypical exemplars to assist with valid and reliable code application.

## Indexing and sorting

Each semi-structured interview (SSI) was used as a unit of analysis. To determine 'what parts of the data are about the same thing and belong together' [31], labels were applied to 'chunks' of data with the same meaning to decide the category/ theme from the framework to assign the text to. Using NVivo, the selected (highlighted) text was 'dragged and dropped' into the relevant sub-themes. This process was followed for all the transcripts.

Three interviewers (AR, FD, and WP) systematically applied the framework to all the transcripts after achieving an inter-rater reliability of 0.87 (Kappa ($\kappa$) value) with the first two transcripts. There were no significant disagreements between the three coders on any subthemes or categories.

## Mapping and interpretation

The finalized themes and subthemes were grouped together. Once the main themes and subthemes were reviewed and finalised, a matrix was created for each theme using Excel, with individual columns for the sub-themes. The first column of the matrix contained case identification details (demographics), followed by summaries of individual themes in subsequent columns. Representative quotes were selected from the SSI to illustrate key themes and subthemes. These themes and subthemes are represented pictorially in Fig 1.

To gain a deeper understanding of the barriers across the demographic groups, we examined the endorsement of the themes across key demographic groups. This included gender (male and female), age groups (less than 40 years versus 40 years and above), and highest educational status attained (tertiary (diploma, degree, and post-graduate education) versus lower than tertiary education (primary, secondary and high school).

All analyses were conducted using Nvivo V.11 (QSR International. NVivo V.11).

## Results

A total of 30 participants (14 females and 16 males) participated in the study. The mean age of participants was 44.7 years (SD = 14.7), ranging from 21 to 75 years (Table 1).

Five main themes pertaining to personal, interpersonal, environmental, socio-cultural, and policy-level factors were classified under the two overarching categories of barriers and facilitators of healthy lifestyles. The personal, interpersonal, environmental, and policy-level factors

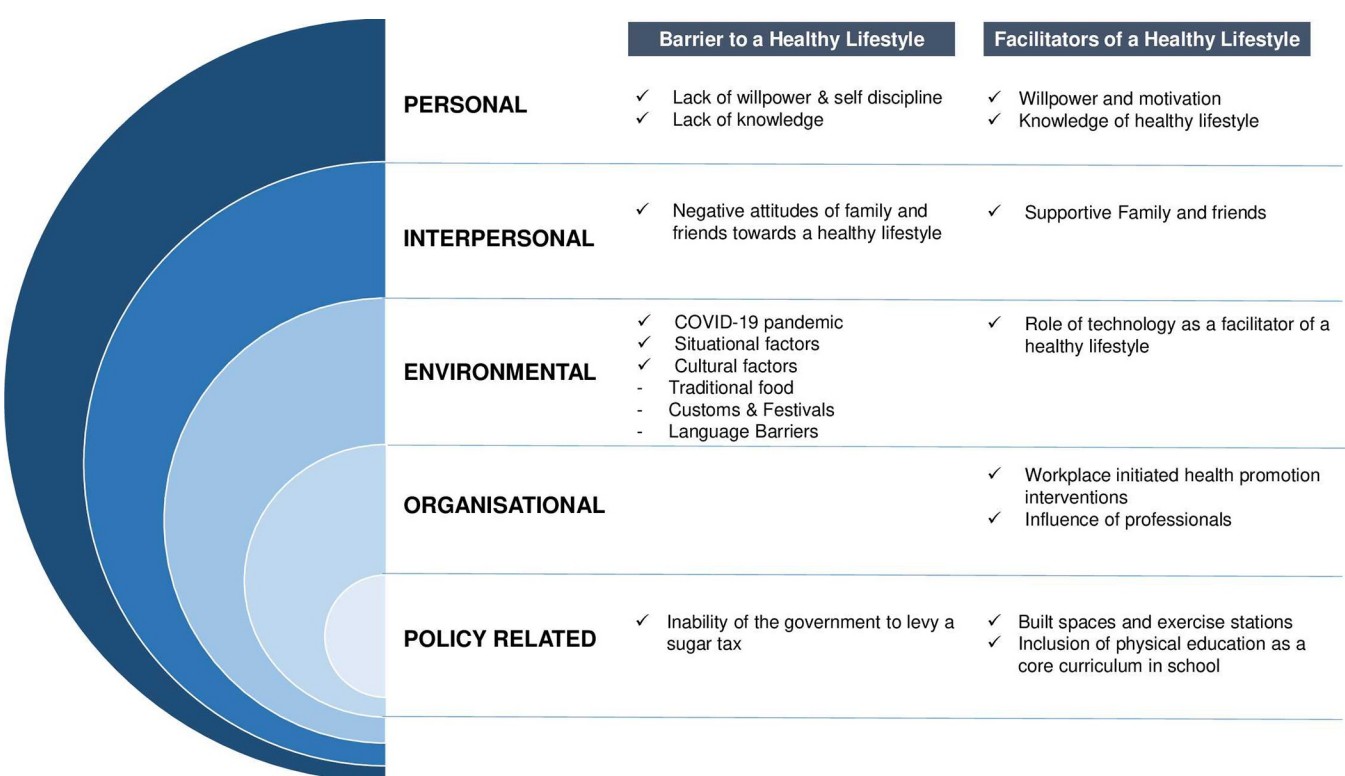

**Fig 1. Barriers and facilitators of adopting a healthy lifestyle.**

comprised subthemes and are highlighted below. Fig 1 shows the summary of the findings of the themes and sub-themes. Minimally edited verbatims that preserve and highlight the participants' experiences and beliefs have been included. In addition, the details of the participants have been provided in brackets as Subject ID/Age/ Gender (**M**ale or **F**emale).

## Barriers to a healthy lifestyle

**Personal factors.** Personal factors comprised two main subthemes explaining the barriers to adopting a healthy lifestyle. These included:

*Lack of willpower and self-discipline.* About one-third of participants mentioned the lack of willpower as an important barrier to maintaining a healthy lifestyle in terms of diet and exercise. They described people as 'being lazy,' using 'tired,' 'too busy,' and 'work' as an excuse not to partake in physical activity. They also alluded to personal dietary preferences such as liking sweets and desserts and people not having the willpower to resist them. A few respondents felt that people were aware of the ill effects of consuming too much sugar, yet they did not have the self-discipline to limit their intake. In addition, respondents felt that people generally knew about the negative outcomes of smoking and alcohol. Yet, they did not quit smoking/ drinking as they did not have the willpower to do it and instead made-up excuses when asked to quit. Respondents also felt that while people may have good intentions and want to adopt a healthy lifestyle, they lacked the willpower to follow through and fell back into unhealthy habits.

> "*It is your own self-cultivation and self-discipline that has to do with your health. If you don't have good self-discipline and you mess around, what kind of healthy body will you have?*" *(SS022/39/M)*

**Table 1. Socio-demographic profile of participants.**

| Subject ID | Age | Ethnicity | Gender | Language of interview | Education | Employment |
|---|---|---|---|---|---|---|
| SS001 | 25 | Chinese | F | English | Diploma | Employed |
| SS002 | 24 | Chinese | M | English | Diploma | Student |
| SS003 | 39 | Malay | F | English | Diploma | Employed |
| SS004 | 38 | Malay | F | English | Post graduate degree | Homemaker |
| SS005 | 54 | Chinese | F | English | Degree | Employed |
| SS006 | 25 | Malay | M | English | Diploma | Employed |
| SS007 | 34 | Malay | M | English | Completed secondary education | Employed |
| SS008 | 27 | Malay | M | English | Degree | Employed |
| SS009 | 31 | Malay | F | English | Degree | Employed |
| SS010 | 22 | Chinese | M | English | Completed high school (equivalent) | Student |
| SS011 | 53 | Chinese | F | English | Completed high school (equivalent) | Employed |
| SS012 | 55 | Malay | F | English | Completed primary education | Employed |
| SS013 | 56 | Chinese | M | English | Diploma | Unemployed |
| SS014 | 26 | Chinese | F | English | Degree | Employed |
| SS015 | 46 | Others | F | English | Post graduate degree | Employed |
| SS016 | 38 | Chinese | M | English | Vocational Institute Training | Unemployed |
| SS017 | 24 | Others | M | English | University degree | Employed |
| SS018 | 58 | Indian | M | English | Completed secondary education | Employed |
| SS019 | 52 | Indian | M | English | Post graduate degree | Employed |
| SS020 | 54 | Indian | M | English | Degree | Employed |
| SS021 | 60 | Chinese | M | Chinese | Some secondary education | Employed |
| SS022 | 39 | Chinese | M | Chinese | Degree | Employed |
| SS023 | 60 | Chinese | M | Chinese | Completed secondary education | Unemployed |
| SS024 | 59 | Chinese | F | Chinese | Completed secondary education | Employed |
| SS025 | 54 | Malay | F | Malay | Completed primary education | Unemployed |
| SS026 | 71 | Malay | M | Malay | Completed primary education | Homemaker |
| SS027 | 65 | Malay | F | Malay | Some secondary education | Homemaker |
| SS028 | 37 | Indian | M | Tamil | Completed secondary education | Employed |
| SS029 | 61 | Indian | F | Tamil | Completed primary education | Employed |
| SS030 | 53 | Indian | F | Tamil | Some secondary education | Homemaker |

"*It's really hard to change. I have been in contact with a few of them (referring to smokers). One of them stopped smoking for three months. "I stopped smoking." After a few days, he started again. I asked him, 'What do you smoke for?' "Oh, pressure." Pressure is fake." (SS021/ 60/M)*

*Lack of knowledge.* Participants felt that it was difficult for someone with a chronic health condition to exercise. They also felt that as a person becomes older, they should avoid vigorous exercise as they could injure themselves more easily. Thus, they thought it was advisable for older people and those with chronic health conditions to reduce their exercise. They did not seem aware that these groups could exercise safely and substitute high-impact activities for lower-impact ones. Regarding diet, respondents expressed frustration with contradictory messages on what was healthy. They felt that food once considered healthy was no longer believed to be healthy and vice versa. Thus, they were unsure of what should be consumed and what should be avoided. A few respondents identified social media as a significant source of unreliable health information.

"*For those who have health conditions, it is very difficult for them to do exercise daily. They are in a life situation where they just cannot take part in a lot of things.*" (SS030/53/F)

"*. . . especially those messages on social media and WhatsApp about your health. Some people say, "Don't take coconut." Some people say, "Yeah. Coconut is healthy. You take more coconut." So really difficult to judge which is right or which is wrong.*" (SS019/52/M)

**Interpersonal factors.**   *Negative attitudes and negative influences of family and friends towards a healthy lifestyle.* Interpersonal factors were mainly identified as the negative attitudes of family and friends towards a healthy lifestyle or the influence of unhealthy practices of friends and family members. E.g., participants mentioned over-eating during family occasions as there was an excess of food and family members or friends urged them to eat more during social gatherings. Participants said that friends who ridiculed their healthy eating habits were barriers to adopting a healthy lifestyle. Several participants shared that when they have food with friends, they tend to over-eat or eat food that is not particularly healthy. They felt uncomfortable not eating the food as they were afraid to be perceived as spoilsports if they did not partake in the feasting and drinking.

"*I think there's a stigma against healthy food. I know how some of my friends say that eating your salad is girly or whatnot. So maybe eating salad is associated with teenage girls, I guess. I don't know. And I guess maybe if they smoke a stick or drink beer, it is more associated with masculine ideals. So maybe if you don't follow the party or whatnot, you may be viewed as an outlier or something. So yeah. I would say there is some social stigma tied to certain types of food or lifestyle, I guess.*" (SS010/22/M)

"*. . .for example, go out with friends and drink milk tea (referring to bubble milk tea, a sweet tea with tapioca balls that is very popular in Singapore) together, and then he drinks a cup and buys you a cup. Will you not drink it? Sometimes, it's not very polite to say no to your friends.*" (SS022/39/M)

**Environmental factors.**   As most of the interviews were conducted during the COVID-19 pandemic when outdoor activity was restricted, it was not surprising that participants mentioned the pandemic as a significant barrier to physical activity. The fear of infection, safe distancing measures, and other restrictions hindered participants from performing outdoor activities and those conducted in gyms or enclosed spaces. The other sub-theme that emerged pertained to situational factors such as conflicting demands leading to time constraints and the low cost and ready availability of fast food that were perceived to be barriers to adopting a healthy lifestyle.

*COVID-19 pandemic.* All the participants mentioned the impact of the COVID-19 pandemic on their lifestyles and expressed their fears and frustrations. For example, participants talked about how the social distancing regulations, closure of indoor gyms and training spaces, and the need to mask up (even in outdoor spaces in Singapore) were significant barriers to exercising in Singapore. Moreover, wearing a mask for most of the day in Singapore's hot and humid weather left them tired, irritable, and reluctant to exercise.

"*Actually, before COVID, my friends and I played soccer on a weekly basis, every Sunday. Yeah. But because of COVID, then we stopped completely.*" (SS008/27/M)

"*And looking at the number of cases (Covid cases), they went up higher. So, it's a deterrence to exercising.*" (SS022/39/M)

*Situational factors.* The most discussed barrier was time constraints associated with competing priorities such as employment, household chores, and looking after children or older parents. However, other factors, such as financial constraints and limited access to healthy food, were also reported to impact healthy lifestyle behaviours.

Participants mentioned that healthy food was both expensive and not readily available. They acknowledged that fast food was the most convenient food, and while they knew that consuming a diet rich in calories was associated with being overweight, they were unable to avoid it. The easy availability of fast food at all hours and food stalls that stayed open even during the night in Singapore also encouraged poor food habits. Interestingly, food delivery was associated with an unhealthy lifestyle. Participants felt that food delivery led people to order more due to convenience and easy availability. In addition to that, it also led to them eating at odd hours.

While most participants acknowledged that Singapore had several parks and exercise areas that were conducive to physical activity, three of the respondents had concerns about the safety of these facilities. These included dimly lit parks that made walking difficult at night and sharing the same path by pedestrians, cyclists, and children and teenagers who tended to run or skate, thus increasing the risk of accidents among older adults. In addition, two participants highlighted the lack of good cycling tracks in Singapore, which does not encourage a cycling culture, unlike Denmark or the Netherlands.

> "*Because sometimes, frankly speaking, children are young, and it is hard for you to have any free time for yourself.*" (SS024/59/F)

> "*I think maybe the fact that healthy food is quite expensive in Singapore. So, I guess food, in general, can be quite affordable if you go hawker centre* (i.e., open-air complexes with many stalls that sell a wide variety of affordably priced food) *or whatnot. But then they are generally not very healthy, so it can be quite troublesome for some people to cook healthier food.*" (SS010/22/M)

> "*Now you can even use your phone to just order, and they will deliver it directly to your house. Ordering food or eating outside food has become so easy that it has become a part of their lifestyle. So, they don't give much consideration to the food itself, and a healthy lifestyle is lost.*" (SS030/53/F)

*Cultural factors.* Several cultural factors emerged as barriers to the adoption of healthy lifestyles. Given the multi-ethnic nature of Singapore, barriers pertaining to cultural factors were identified both by people belonging to that ethnocultural group and others. These included:

*The cultural importance of traditional food.* Participants acknowledged that cooking and eating traditional food was an important ritual in Singapore. However, about one-third of the participants felt that Indian and Malay food tended to be oily and calorie-rich. They also acknowledged that these types of food appealed to people's taste and they ended up overeating them. In addition, they felt that desserts unique to these cultures were similarly sweet and not healthy. Coconut milk in traditional food preparations was similarly identified as an unhealthy but necessary ingredient. Some also commented that those of the Chinese ethnicity liked to eat pork and were unwilling to switch to healthier meat alternatives (such as white meat). They also acknowledged that Chinese cuisine could be oily as many dishes are deep-fried.

> "*In fact, Singapore's food is not only western food, but also Malay food, Indian food, and Chinese food. It doesn't contain as much oil as Chinese cuisine in China. Chinese food has a lot of oil. The Chinese food here, oil and salt, will not be so overused, but it will have a lot of fried*

*things. And your Indian food and Malay food, I believe it, will have a lot of sugar, especially Malay food." (SS022/39/M)*

*Customs and Festivals.* Participants also talked about the food habits of specific ethnic groups, such as eating dinner late at night and close to bedtime, which they perceived as unhealthy. They also acknowledged that festive periods were not conducive to maintaining a healthy lifestyle as it was all about meeting friends and families and eating. So, one tended to overeat during such periods.

*"As far as we Chinese are concerned, if we talk about the Chinese New year, it may be that everyone eats more. . ."* (SS021/60/M)

*Language barriers.* People from minority ethnicities expressed their reluctance to participate in community-based group exercise programs as they felt they would not be able to understand the instructions as these tend to be conducted in a language, they are not conversant with.

*". . .that's why I just hate to go to some of these community activities. They are all in English, and I won't understand. Most or all are in English only, so I feel a little uncomfortable because of that."* (SS030/50/F)

**Policy related factors.** Policy-related barriers did not emerge very strongly in this group of participants. However, a few participants expressed their frustration with the reluctance of the government to impose a sugar tax. They felt that sugar caused significant harm to a person's health, but it was not something that could be taxed. They mentioned the 'bubble tea fad' in Singapore, leading to several shops selling sweet and calorie-rich drinks across the country. The existing policies could not limit such shops; the government, they felt, could only advocate and educate people about the potential harms of such food.

*"I don't think it's realistic. Because, for example, the government tells people not to smoke, then they increase the tax on cigarettes. Don't drink, and they will add a high tax to wine. But it is impossible to add a high tax on sugar because sugar is a necessity in life. Unlike tobacco and wine, it is not a luxury but a part of the diet. Many people in their daily life use sugar. So, you can't, the government can't say, add a high tax to milk tea shops. So, I think from the government's point of view that they can't do many things. They can just advocate."* (SS022/39/M)

Analysis of barriers across the socio-demographic groups revealed differences in the endorsement of sub-themes. Those who were older, i.e., 40 years and above, endorsed a lack of willpower and self-discipline as a barrier to adopting a healthy lifestyle. Women were more likely to endorse situational factors and customs and festivals as barriers. In contrast, more men endorsed the cultural importance of local food as a barrier to a healthy lifestyle. Those with a tertiary education did not feel that language was a barrier to participating in activities, and only four of them endorsed a lack of willpower and self-discipline as a barrier to the adoption of a healthy lifestyle (Table 2).

### Facilitators of a healthy lifestyle

**Personal and interpersonal factors.** Most of the facilitators highlighted by the participants in these two themes were the opposite of those mentioned as barriers. However, the

**Table 2. Barriers and facilitators endorsed by respondents across socio-demographic groups discussion.**

| | Male (n = 16) | Female (n = 14) | < 40 years (n = 14) | ≥40 years (n = 16) | Tertiary education (n = 15) | Less than tertiary education (n = 15) |
|---|---|---|---|---|---|---|
| **Barriers** | | | | | | |
| *Lack of willpower and self-discipline* | 7 | 6 | 4 | 9 | 4 | 9 |
| *Situational factors* | 7 | 9 | 8 | 8 | 8 | 8 |
| *The cultural importance of traditional food* | 7 | 4 | 5 | 6 | 5 | 6 |
| *Customs and Festivals* | 3 | 6 | 4 | 5 | 4 | 5 |
| *Language Barriers* | | 2 | | 2 | | 2 |
| **Facilitators** | | | | | | |
| *Workplace-initiated health promotion interventions* | 6 | 6 | 11 | 1 | 10 | 2 |
| *Built spaces and workstations* | 10 | 7 | 9 | 8 | 6 | 11 |

*Only subthemes that were different across socio-demographic factors have been highlighted

Tertiary includes diploma, degree and postgraduate education

absence of a barrier was not necessarily a facilitator. For example, most participants highlighted 'willpower and motivation' as personal facilitators. Participants talked about how willpower was necessary to exercise regularly and eat healthy food. They also felt that if people knew the impact of a healthy lifestyle on long-term outcomes, they would commit to them. One-third of participants also acknowledged that people with health conditions should continue to exercise and maintain a healthy diet as it can prevent secondary complications. They also highlighted the important role of friends and family members in encouraging and supporting a healthy lifestyle, which helped the participants maintain it.

**Organisational/ institutional factors.** *Workplace-initiated health promotion interventions.* The workplace emerged as a significant facilitator of a healthy lifestyle. More than half of the participants who were employed mentioned various workplace initiatives that had helped them to become more physically active. This was mainly through workplace wellbeing initiatives such as the distribution of fruits, subsidized fruit bazaars at the place of work, educational sessions on diet and its impact on well-being, and group exercise classes like Zumba or Yoga. Many workplaces continued these initiatives even during the pandemic by leveraging Zoom and other platforms.

*"But my workplace, I would say they are trying to endorse the whole healthy lifestyle thing. So, we do have things like a monthly fruit giveaway. So, every single month we'll get different kinds of fruit and then staff will explain to us the benefit of eating food, or fruit rather. We have staff exercise sessions where you can sign up for yoga or gym sessions or go for a walk."* (SS009/31/F)

*Influence of healthcare and other professionals.* Several participants talked about adopting a healthier lifestyle after their healthcare provider (usually a doctor or dietician) advised them about healthy eating or physical activity. They also spoke of informative media programs that encouraged the adoption of a healthy lifestyle.

*"Polyclinics (primary care clinics) or I think if I didn't recall wrongly—I can't really remember. Is it one of the hospitals or polyclinics my parents visited? They actually have a nutritionist who tells you what to eat."* (SS016/38M)

*"White meat, the doctor's advice is to eat more white meat instead of red meat because red meat is not good for cholesterol."* (SS022/39/M)

**Environmental factors.** *Role of Technology as a facilitator of a healthy lifestyle*. All the participants highlighted the role of technology as a facilitator of a healthy lifestyle. However, their understanding of technology varied. Any source of information like television, radio, Internet search engines, channels like YouTube, social media sites, and Apps (mobile applications) was described as technology. Participants saw technology as enabling access to information on diet and exercise, aiding in training and tracking and monitoring their physical activity, heart rate, sleep, and food consumption. Some participants alluded to Apps that sent reminders to breathe deeply, meditate, and walk as helpful.

"*App for cycling that's called Strava. I think that's just the only healthy fitness app that I have. So, they will keep track of your heartbeat, the distance from one point to another, and the speed of cycling.*" (SS007/34/M)

"*It's MyFitnessPal. Well, it tracks my calorie intake for the day. And it's quite specific. It's quite good. But I think they cannot detect some of the local foods. But other than that, they can track basically whatever that goes in your mouth, yeah, whatever you consume. And you can put in your exercises for the day. So, yes, it will help you calculate your goals, like how many KG you want to lose in a month also, which is quite good.*" (SS008/24/M)

**Policy-related factors.** The participants highlighted several policy-related facilitators. These included:

*Built spaces and exercise stations*. More than half of the participants talked about the availability of neighbourhood parks which provided a safe and convenient place to exercise. Participants mentioned that grassroots organisations often organised walks in their neighbourhood and that volunteers would encourage them to join in these activities. They were also aware that several activities were conducted in these spaces that one could join at no cost. Some also felt that such group activities motivated them, and they enjoyed doing these more than doing exercises by themselves. Participants also mentioned that there are public swimming pools that one could use and government-run gyms where one could access high-quality equipment at a minimal cost. Participants acknowledged that the government was constantly upgrading parks and gyms, and they could now easily access parks and exercise corners.

"*Walking, our government is good, gave us many parks and so many connectors (scenic roads connecting parks where pedestrians can walk). You cannot say that there are no facilities.*" (SS024/59/F)

"*Sometimes they do come and call me to join, like what's that called, social service Community volunteers. When we exercise with other people in a group, it gives us a sort of motivation.*" (SS030/53/F)

*Inclusion of physical education as a core curriculum in school*. About a quarter of the participants mentioned that including physical education in schools and getting children to exercise regularly as part of the school curriculum encouraged incorporating exercise into their lifestyle. They also shared that even tertiary education institutes offered an excellent array of exercise classes/ options, which enabled the students to continue exercising.

"*The education system right now really talks about mental health and physical health. Even their physical education is different. So, I think exposure to that will be one of the factors that will enable them to live a healthy lifestyle.*" (SS009/31/F)

*"So, when I went to XXX Poly [polytechnic name], I started Muay Thai. And even the gym was affiliated with XXX [polytechnic name]. So, I've been with this gym since the dawn of time. This is my first gym, and I've been with them all the way [laughter]." (SS006/25/F)*

Analysis of facilitators across the socio-demographic groups revealed differences in the endorsement of sub-themes. Those younger, i.e., below 40 years of age and with tertiary education, endorsed workplace-initiated health promotion as a facilitator more than those aged 40 years and above and with lower education. Those with a lower than tertiary education endorsed built spaces and workstations as a facilitator more than those with tertiary education (Table 2).

## Discussion

This article explored the barriers and facilitators of a healthy lifestyle perceived and experienced by a multi-ethnic sample of adults in Singapore. Using a framework analysis approach that comprised two major components: creating an analytic framework and applying this analytic framework, we leveraged the SEM [22] model to gain a deeper understanding of the barriers and facilitators of a healthy lifestyle. The discussion focuses on key themes that lend well to intervention or were unique to this study.

At the personal level, lack of willpower emerged as a key barrier, while being motivated and having the resolve to exercise or not eat sweet or calorie-rich food, despite the challenges, was identified as a facilitator. Willpower, defined as the capacity to exert self-control, has emerged in several studies as a barrier to healthy eating [32, 33] and physical activity [34]. In Tsukayama et al.'s [35] prospective longitudinal study, the researchers found that more self-controlled children were less likely to become overweight as they entered adolescence. Cognitive and behavioural interventions have been developed to promote self-regulation [36] and overcome this barrier. For example, a study among women aged 30–50 showed that a brief intervention combining information with a self-regulation technique led to the maintenance of high consumption of fruits and vegetables 24 months after the intervention. In contrast, the information-only intervention group returned to baseline consumption of fruits and vegetables [37].

Similarly, another study tested an intervention that combined information with cognitive-behavioural strategies on women's physical activity with an information-only intervention. The women who were randomly assigned to the self-regulation and information session were substantially more active than those who participated in the information-only sessions [38]. While the WoD campaign has ensured the dissemination of information on a healthy lifestyle, there is a need to develop and evaluate interventions that provide information and teach and promote self-regulation. If such interventions are effective, they could be scaled up at the population level.

Environmental factors that emerged as barriers and facilitators were unique to this study. The study period coincided with the COVID-19 pandemic, and all the participants mentioned the pandemic as a barrier to adopting a healthy lifestyle. While some participants spoke about the importance of their own 'willpower' in maintaining their exercise regimen during the pandemic, they acknowledged the challenge posed by the pandemic. Other studies have similarly reported dramatic lifestyle changes in reducing physical activity with increased sedentary behaviours and reduced physical activity during the Covid-19 pandemic [39, 40]. These unhealthy lifestyle behaviours observed in the pandemic can potentially lead to the persistence of these poor lifestyle habits and the development of chronic diseases. Contact tracing mobile Apps were rolled out very early during the COVID-19 pandemic, in Singapore, mainly as a means of infection control [41]. However, at the national level, there was no imperative to

develop or implement apps that could provide information on healthy lifestyles during the period of enforced social isolation. A conversational agent such as Elena+ [42], which provided coaching sessions, behavior change activities, and intention/goal formation to promote a healthy lifestyle during the pandemic, must be culturally adapted and implemented globally. Such digital health interventions are of value both during the pandemic and beyond it to ensure that at-risk populations are engaged in health promotion [42].

This study identified several cultural factors as barriers to adopting a healthy lifestyle. The prevalence of diabetes varies among Chinese, Malay, and Indian ethnicities, and it is often ascribed to dietary differences, especially in the popular media. Interestingly, while some participants of Indian and Malay ethnicities said that their food choices and food preparation might be high in calories, many Chinese participants also commented that traditional Malay or Indian food was too sweet or oily. However, most of our participants acknowledged cross-cultural eating and said they preferred deep-fried or sweet food. They also felt that they ended up eating more of it than low-fat options that were often not spicy or tasty. All the major festivals celebrated in Singapore were identified as periods where people choose not to count calories and enjoy feasting with friends and families, highlighting the importance of traditional food during social gatherings and religious or traditional celebrations. A study on South Asian immigrants in Australia identified a similar theme where participants felt that food was a central theme of social gatherings and indicated their preference for traditional food in these settings [43]. Multilevel interventions targeted towards families, i.e., those that involve children and parents, comprising programs that increase knowledge, willingness to try nutritious food, and encourage menu modifications without compromising on taste, could be trialed in Singapore [44]. Given the importance of family in Asia and the role of interpersonal factors as both barriers and facilitators in this population, such an approach may be both appealing and strategic.

Workplace and technology emerged as significant environmental facilitators. The Health Promotion Board, Singapore, has spearheaded workplace initiatives. They identified workplaces as a critical setting as most adult Singapore residents spend most of their day at work. The focus areas include obesity prevention and management and chronic disease management. They work proactively with companies to support them with the necessary tools to ensure a health-promoting workplace. These initiatives have resulted in many companies providing workplace talks on physical and mental health, organising group physical activities, providing healthy food alternatives in canteens, subsidising the cost of fruits, and distributing fruits and healthy snacks to staff [45]. Technology has been classified as a component of the physical environment's artificial elements [46]. Given the focus on developing Singapore as a 'smart nation' to leverage technology and implement it nationally, the widespread interest and adoption of technology for a healthy lifestyle were not surprising. With the rapid technological advances and integration of smartphones with wearable devices that can assess physical activity, sedentary behavior, heart rate, and intensity levels of physical activity [47], many people prefer wearables as facilitators of a healthy lifestyle.

Furthermore, technological advancement has resulted in better identification and tracking of previously non-identifiable physical activity (e.g., stair climbing, outdoor cycling), which our participants mentioned as particularly appealing as they catered to their lifestyle. Participants also alluded to the persuasive technology that was often incorporated within the wearables. The ability to send their achievements (hours exercised, distance covered, etc.) to online communities or friends with whom they could compete, stay accountable or get encouragement for their achievements was seen as a facilitator by some participants. At the same time, others mentioned the reminders to pause and take deep breaths as being useful in the middle of stress-filled days. Persuasive technology is defined as technology that is designed to change

individuals' attitudes or behaviours through persuasion and social influence, but not through coercion [48]. Technological advances can be used to nudge individuals to engage in more physical activity. This is done by capturing the data and comparing it with historical data; tracking improvement over time; linking data to social media; and sending encouraging messages to the wearer, such as asking them to move more or taking some time for deep breathing or mindfulness. While these are exciting and transformative developments that facilitate the adoption of a healthy lifestyle, little data is available concerning how successful these apps or wearables are at enabling users to lose weight or get fit over time. Research into the effectiveness of many of these technologies is still in its infancy.

Policy-driven changes to the built environment and education curriculum were identified by participants as facilitators of the adoption of a healthy lifestyle. Our study highlights the importance of positioning health promotion in city planning and developing 'healthy built environments.' Surprisingly, participants did not mention the 'utilitarian walking' encouraged in Singapore due to greater land-use mixes that ensure easy accessibility to various locations in the neighbourhood, like shops, food courts, and primary care services. Instead, they focused mainly on recreational walking enabled by small open places like playgrounds and large parks within walking distance of the residential neighbourhoods. Many pointed out that with other walkability features such as safe sidewalks, covered pathways, and easy accessibility, there was no excuse not to exercise in Singapore. Several other studies in Singapore have similarly established that physical activity levels are closely associated with the built environment characteristics [49, 50].

The incorporation of physical education classes in the school curriculum meets several important objectives. First, it ensures that students participate in appropriate amounts of physical activity during lessons. Secondly, they become equipped with the knowledge and skills to be physically active throughout life [51]. Several participants referred to the need to incorporate physical activities into the routine right from childhood as they felt that children who learn these skills would use them lifelong. And that it is more difficult to convince older adults to do physical activities, especially if they have not done them before. However, schools should consider providing a diverse range of physical activity experiences so that the needs and interests of all children are met. Schools should also consider incorporating healthy eating practices as part of their curriculum to reduce the risk of childhood obesity further and promote lifelong healthy nutritional practices [52].

There are some limitations to our study. Since the study was planned before the COVID-19 pandemic, in the early part of the study, the team did not have specific questions or probes that examined factors that could be related to the pandemic. The pandemic may have also limited those who were not technologically savvy or worried about the impact of the infection from participating in the study. The pandemic experience may have coloured the participants' attitudes towards healthy lifestyles, i.e., they may have had a more positive attitude toward it, given the higher risk of poor Covid-19 infections amongst those with multimorbidity and other lifestyle-associated risk factors. While the study allowed the participants to define what a healthy lifestyle meant to them, the discussion on barriers and facilitators centred mainly around physical activity and nutrition. There was limited discussion around other aspects of a healthy lifestyle, like using tobacco, alcohol, sleep, and mental health. The authors could not link the quantitative and qualitative data due to ethical considerations. During the qualitative phase, we did not collect such data (BMI, smoking habits, etc.), limiting a deeper understanding of the participant's narratives. The strengths of our study include a good representation of people across ethnic groups and languages in a multi-ethnic population. The use of one-to-one interviews that led to frank discussions on barriers and facilitators and the inclusion of data from 30 interviews ensured thematic saturation. The qualitative researchers involved in this

study came from different disciplines, thus providing a transdisciplinary understanding of the phenomenon under study. Lastly, the study results triangulate well with other studies examining barriers and facilitators of healthy lifestyles.

## Conclusions

Our study found that participants were aware of the several steps undertaken by the Singapore Government to promote a healthy lifestyle. The school-based and workplace health-promoting activities were seen as promoting and ensuring the adoption of a healthy lifestyle. Participants were also cognisant of the built environment in Singapore that encourages adopting a healthy lifestyle. However, despite these consistent efforts by the Singapore Government, participants identified several barriers to adopting a healthy lifestyle. Personal and interpersonal factors like willpower, self-regulation, and influence from family and peers were identified as important barriers. On the other hand, devices for monitoring activities and diet emerged as significant facilitators that can be further leveraged to improve the health of populations.

Most of the barriers identified are amenable to interventions. Incorporating educational material with motivational techniques, short interventions to improve self-regulation delivered by health care professions, multilevel interventions targeted at families, and nudge technology to promote a healthy lifestyle should be explored in future studies.

## Supporting information

**S1 File. Semi-structured interview guide.**
(DOCX)

## Author Contributions

**Conceptualization:** Mythily Subramaniam, Janhavi Ajit Vaingankar, Siow Ann Chong.

**Formal analysis:** Mythily Subramaniam, Fiona Devi, P. V. AshaRani, Peizhi Wang, Anitha Jeyagurunathan.

**Funding acquisition:** Mythily Subramaniam, Siow Ann Chong.

**Investigation:** Mythily Subramaniam, Siow Ann Chong.

**Methodology:** Mythily Subramaniam, Fiona Devi, P. V. AshaRani, Yunjue Zhang, Peizhi Wang, Anitha Jeyagurunathan, Kumarasan Roystonn, Janhavi Ajit Vaingankar, Siow Ann Chong.

**Project administration:** Mythily Subramaniam, Fiona Devi, P. V. AshaRani, Yunjue Zhang, Kumarasan Roystonn, Janhavi Ajit Vaingankar.

**Resources:** Mythily Subramaniam, Siow Ann Chong.

**Supervision:** Mythily Subramaniam, Siow Ann Chong.

**Writing – original draft:** Mythily Subramaniam, Fiona Devi.

**Writing – review & editing:** Fiona Devi, P. V. AshaRani, Yunjue Zhang, Peizhi Wang, Anitha Jeyagurunathan, Kumarasan Roystonn, Janhavi Ajit Vaingankar, Siow Ann Chong.

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
