## [Decision Letter · Decision Letter 0]

1 Aug 2022

PONE-D-22-13181Barriers and Facilitators for adopting a healthy lifestyle in a multi-ethnic population: A qualitative studyPLOS ONE

Dear Dr. Mythily Subramaniam,

Thank you for submitting your manuscript to PLOS ONE. After careful consideration, we feel that it has merit but does not fully meet PLOS ONE’s publication criteria as it currently stands. Therefore, we invite you to submit a revised version of the manuscript that addresses the points raised during the review process.

ACADEMIC EDITOR:

The topic of this paper is of interest to the Plos One scientific community. Adherence to healthy lifestyle behaviours remains a challenge, thus exploring which factors might facilitate or hinder the adoption of these behaviours is critical to improve intervention’s design and efficacy. Still, the current paper does not show the novelty of this study. This should be made totally clear in all sections of the paper, but most importantly in the introduction and discussion sections. Authors are also advised to be more specific and support all affirmations in the literature.

The methodology should be clearer, especially regarding the process of development of the instrument and how the framework analysis was conducted.

The impact of COVID on recruitment and on this study should be discussed, as these results could be different in a normal, non-COVID, situation. Being the first study on the Singapore population, this should be interpreted with caution and unlikely generalisable. Therefore, avoid using terms that imply that your findings are comprehensive and generalisable to all Singapore population within the same age range.

Grammatical errors should be carefully revised.

The authors should pay close attention to the reviewers’ comments, carefully addressing all of them.

Please submit your revised manuscript by Sep 15 2022 11:59PM. If you will need more time than this to complete your revisions, please reply to this message or contact the journal office at plosone@plos.org. Please include the following items when submitting your revised manuscript:

We look forward to receiving your revised manuscript.

Kind regards,

Eliana Carraça

Academic Editor

PLOS ONE

Journal Requirements:

“This study is supported by the Singapore Ministry of Health’s National Medical Research Council under its Health Services Research Grant (NMRC/HSRG/0085/2018).”

“SAC received the funding

NMRC/HSRG/0085/2018

This study is supported by the Singapore Ministry of Health’s National Medical Research Council under its Health Services Research Grant

https://www.nmrc.gov.sg/who-we-are

Reviewers' comments:

Reviewer's Responses to Questions

**Comments to the Author**

1. Is the manuscript technically sound, and do the data support the conclusions?

Reviewer #1: Yes

Reviewer #2: Yes

2. Has the statistical analysis been performed appropriately and rigorously? 

Reviewer #1: N/A

Reviewer #2: N/A

3. Have the authors made all data underlying the findings in their manuscript fully available?

Reviewer #1: Yes

Reviewer #2: No

4. Is the manuscript presented in an intelligible fashion and written in standard English?

Reviewer #1: Yes

Reviewer #2: Yes

5. Review Comments to the Author

Reviewer #1: This paper provides interesting insight into the barriers and facilitators to adopting healthy lifestyles among adults in Singapore. The paper is clear and well written and I appreciate the inclusion of the interview guide. I would suggest the following revisions to improve the manuscript:

1) As the authors note, many studies have been published describing barriers and facilitators to healthy lifestyle practices. While this study is novel in its focus on Singapore, the results do not seem to differ much at all from the large body of work on this topic. It would be useful in the discussion to frame how the findings of this study compare to other similar studies. Are there barriers and facilitators specific to adults in Singapore that could be targeting in a healthy lifestyle intervention?

2) Authors make the claim that this study provides a "comprehensive" understanding of barriers and facilitators (pg 5). I am not sure this claim can be made with a fairly small sample (n=30). The paper can still bring forward important findings without being comprehensive.

3) Pg 8: spell out "SSI" on first use

4) Pg 8, table 1: Given the important of education level and adoption of healthy lifestyle habits (as the authors mention in the introduction), it's surprising that education is not included in the demographics. It would also be interesting to see how barriers and facilitators differed among those with varying levels of education.

5) Similarly to the comment above, it would have been nice to see a segmentation of the data by demographics. For example, are barriers and facilitators different for men and women? Older and younger people? By ethnicity and/or language?

6) pg 14: What is meant by "hawker center" in the quote? Provide definition or use a different quote.

7) pg 16: What is meant by "community exercises"? Is this physical activity opportunities in the community?

8) Informed consent was collected from all participants. Was the study protocol reviewed and approved by an institutional review board?

9) In the introduction, the authors state that this paper will contribute to developing interventions to promote healthy lifestyles among people in Singapore, where diabetes and other chronic conditions are on the rise. I would therefore expect to see explicit recommendations in the discussion or conclusion for interventions tailored to this context. Most of the recommendations mentioned (PA/healthy eating in schools, built environment to promote PA) have been well studied and documented in other contexts and are not novel or innovative. In the revision, I suggest the authors consider what this data adds to the literature and how it can be used to inform intervention development.

10) Various grammatical errors were noted throughout the manuscript (see comments in attached PDF). Suggest careful re-reading and editing for clarity.

Reviewer #2: Please see the review comments in the PDF document. I have highlighted the comments.

Overall, the paper does not show the novelty of this study, even though (as argued but not substantiated) that Singapore has yet to be the focus of such a study.

If authors proceed with a revision, care and clarification is necessary with the manuscript revision to demonstrate study rigor and allow for replication. Further, authors should revisit the discussion and conclusion to 1) enhance the limitations section, and 2) truly demonstrate why this study is novel and important in PLOS One

6. PLOS authors have the option to publish the peer review history of their article (what does this mean?). If published, this will include your full peer review and any attached files.

Reviewer #1: No

Reviewer #2: No

---

## [Author Response · Author response to Decision Letter 0]

26 Aug 2022

27 August 2022

Eliana Carraça

Academic Editor

PLOS ONE

Ref: PONE-D-22-13181

Barriers and Facilitators for adopting a healthy lifestyle in a multi-ethnic population: A qualitative study

Dear Dr. Carraca

We would like to thank you and the reviewers for your constructive review. We have addressed the points raised by the reviewers in the revised manuscript in as tracked changes. Our replies to their comments are attached and highlighted in bold for easy reference. The major comments on the pdf have been addressed separately as we are not sure which reviewer has commented on them. The minor comments have been addressed directly in the text as tracked changes.

Academic Editor

The topic of this paper is of interest to the Plos One scientific community. Adherence to healthy lifestyle behaviours remains a challenge, thus exploring which factors might facilitate or hinder the adoption of these behaviours is critical to improve intervention’s design and efficacy. Still, the current paper does not show the novelty of this study. This should be made totally clear in all sections of the paper, but most importantly in the introduction and discussion sections. Authors are also advised to be more specific and support all affirmations in the literature.

We have added relevant references and clarified the novel aspects of this study.

The methodology should be clearer, especially regarding the process of development of the instrument and how the framework analysis was conducted.

The methodology has been substantially revised as suggested by the reviewers.

The impact of COVID on recruitment and on this study should be discussed, as these results could be different in a normal, non-COVID, situation. Being the first study on the Singapore population, this should be interpreted with caution and unlikely generalisable. Therefore, avoid using terms that imply that your findings are comprehensive and generalisable to all Singapore population within the same age range.

We have avoided the use of terms that suggest that the findings are comprehensive and generalizable.

Grammatical errors should be carefully revised.

Grammatical errors have been revised.

Journal Requirements

“This study is supported by the Singapore Ministry of Health’s National Medical Research Council under its Health Services Research Grant (NMRC/HSRG/0085/2018).”

“SAC received the funding

NMRC/HSRG/0085/2018

This study is supported by the Singapore Ministry of Health’s National Medical Research Council under its Health Services Research Grant

We would like to retain the statement as it is in the online submission form. We have removed the statement in the revised manuscript.

3. In your Data Availability statement, you have not specified where the minimal data set underlying the results described in your manuscript can be found. PLOS defines a study's minimal data set as the underlying data used to reach the conclusions drawn in the manuscript and any additional data required to replicate the reported study findings in their entirety. All PLOS journals require that the minimal data set be made fully available. For more information about our data policy, please see.

Upon re-submitting your revised manuscript, please upload your study’s minimal underlying data set as either Supporting Information files or to a stable, public repository and include the relevant URLs, DOIs, or accession numbers within your revised cover letter. For a list of acceptable repositories, please see. Any potentially identifying patient information must be fully anonymized.

Important: If there are ethical or legal restrictions to sharing your data publicly, please explain these restrictions in detail. Please see our guidelines for more information on what we consider unacceptable restrictions to publicly sharing data: https://imsva91-ctp.trendmicro.com:443/wis/clicktime/v1/query?url=http%3a%2f%2fjournals.plos.org%2fplosone%2fs%2fdata%2davailability%23loc%2dunacceptable%2ddata%2daccess%2drestrictions&umid=C1935020-E531-7305-AE14-9D67384FA132&auth=6e3fe59570831a389716849e93b5d483c90c3fe4-a54b28680ec209732c5562257d5ed9d73bc73984. Note that it is not acceptable for the authors to be the sole named individuals responsible for ensuring data access.

Revised data availability statement: The data underlying the results presented in the study are available from the first author Dr Mythily Subramaniam (Mythily@imh.com.sg)

Reviewer: 1

1) As the authors note, many studies have been published describing barriers and facilitators to healthy lifestyle practices. While this study is novel in its focus on Singapore, the results do not seem to differ much at all from the large body of work on this topic. It would be useful in the discussion to frame how the findings of this study compare to other similar studies. Are there barriers and facilitators specific to adults in Singapore that could be targeting in a healthy lifestyle intervention?

We would like to thank the reviewer for this suggestion. We have revised our introduction to address how the study is different from others that have been carried out in other countries.

2) Authors make the claim that this study provides a "comprehensive" understanding of barriers and facilitators (pg 5). I am not sure this claim can be made with a fairly small sample (n=30). The paper can still bring forward important findings without being comprehensive.

We have removed the word ‘comprehensive’.

3) Pg 8: spell out "SSI" on first use

We apologise for the error and have added the full form as suggested by the reviewer.

4) Pg 8, table 1: Given the important of education level and adoption of healthy lifestyle habits (as the authors mention in the introduction), it's surprising that education is not included in the demographics. It would also be interesting to see how barriers and facilitators differed among those with varying levels of education.

We have included the educational backgrounds of the participants in the revised table.

5) Similarly to the comment above, it would have been nice to see a segmentation of the data by demographics. For example, are barriers and facilitators different for men and women? Older and younger people? By ethnicity and/or language?

We are unable to do such an analysis as the study design was set up to get a good representation across the groups but did not set out to examine differences across the groups. The sample size does not lend to such a nuanced segmentation and we apologise for that.

6) pg 14: What is meant by "hawker center" in the quote? Provide definition or use a different quote.

We have explained what is meant by a ‘hawker center’. We think it is important to retain this quote as it highlights the availability of affordable food which is often not healthy.

7) pg 16: What is meant by "community exercises"? Is this physical activity opportunities in the community?

We have clarified the term by changing it to community-based group exercise programs for clarity.

8) Informed consent was collected from all participants. Was the study protocol reviewed and approved by an institutional review board?

Yes, we have clearly stated just below the statement on consent that - ethical approval for the study was obtained from the relevant institutional review board (National Healthcare Group Domain Specific Review Board; protocol ref:2019/00926).

9) In the introduction, the authors state that this paper will contribute to developing interventions to promote healthy lifestyles among people in Singapore, where diabetes and other chronic conditions are on the rise. I would therefore expect to see explicit recommendations in the discussion or conclusion for interventions tailored to this context. Most of the recommendations mentioned (PA/healthy eating in schools, built environment to promote PA) have been well studied and documented in other contexts and are not novel or innovative. In the revision, I suggest the authors consider what this data adds to the literature and how it can be used to inform intervention development.

We have revised our discussion and conclusion as suggested by the reviewer.

10) Various grammatical errors were noted throughout the manuscript (see comments in attached PDF). Suggest careful re-reading and editing for clarity.

We have revised the manuscript as suggested by the reviewer.

Reviewer #2: 

Please see the review comments in the PDF document. I have highlighted the comments.

Main comments in the pdf 

Introduction

1. Authors should take a paragraph or two to summarize the studies in Asia populations. This would help to demonstrate why a specifically cultural focus/regional focus is important. 

 We have added a paragraph focusing on Asian studies in the revised manuscript.

2. At some point, authors need to explain why the study reported here is focused on general health behaviors vs. Diabetes related health behaviors, as the parent study is a diabetes study. Otherwise, authors are encouraged to situate these findings within health behaviors around Diabetes.

 As we have stated in the revised manuscript the intent of the study was to examine barriers and faciltators to healthy lifestyle in the broader context of general health behaviors which have been promoted at the policy level in Singapore. While the larger study was towards knowledge, attitudes and practices towards diabetes, the questions on lifestyle were general and not towards diabetes related health behaviors (Koh YS, Asharani PV, Devi F, Roystonn K, Wang P, Vaingankar JA, Abdin E, Sum CF, Lee ES, Müller-Riemenschneider F, Chong SA, Subramaniam M. A cross-sectional study on the perceived barriers to physical activity and their associations with domain-specific physical activity and sedentary behaviour. BMC Public Health. 2022 May 26;22(1):1051. doi: 10.1186/s12889-022-13431-2). This was also in consultation with diabetologists who opined that diabetes was often associated with other comorbid conditions and risk factors for several NCIDs were common. Thus, it was decided to understand general health behaviors. We have clarified this in the methodology.

Methodology

3. Please clarify: was this sampling method used for the parent study AND the interview component or both or just the interview component? If a specific sampling method was conducted (were there only 30 people who agreed to be recontacted from the survey? Probably not). So how did authors select FOR RECRUITMENT those who said they’d be willing to be contacted.

 We have clarified the sampling method for the parent study. For the current study the participants were sampled randomly from those who gave consent for recontact. We have clarified the process in the methodology.

4. Are authors claiming that there were no a priori coding structures? If so, why use framework analysis?

 We would like to respectfully state that a framework analysis can be used without apriori coding structures. We have further expanded the steps to explain what we did. While our initial coding was guided by our research questions and hence the topic guide, we did not use apriori coding.

 Framework analysis in applied research have varied from highly deductive analysis (Pope et al., 2000) to inductively-oriented approaches in healthcare research (e.g., Goldsmith et al., 2017; Swallow et al., 2011).

Results

5. Specify cis, trans, nonbinary gender participants. 

 None of the participants identified themselves as cis/ trans or non-binary.

6. Authors should describe more about the participants in this sample: rurality, occupation (or employment), wealth or income, some measure of health (healthy weight? Health behavior such as smoking?). Readers need to understand more about the sample for this interview cohort.

 We have described the sample in Table 1. Singapore is a highly urbanised city-state, there is no rural area in Singapore. We did not link health data to the qualitative interviews, and we have acknowledged it as a limitation of the study. We have added education and employment data.

Overall, the paper does not show the novelty of this study, even though (as argued but not substantiated) that Singapore has yet to be the focus of such a study.

We have revised our introduction to substantiate the novelty of the study.

If authors proceed with a revision, care and clarification is necessary with the manuscript revision to demonstrate study rigor and allow for replication. Further, authors should revisit the discussion and conclusion to 1) enhance the limitations section, and 2) truly demonstrate why this study is novel and important in PLOS One

We have made the changes suggested by the reviewers.

We hope that we have addressed the reviewers’ comments adequately and we look forward to a favourable reply.

Regards

Authors

---

## [Editor Report · Decision Letter 1]

2 Oct 2022

PONE-D-22-13181R1Barriers and Facilitators for adopting a healthy lifestyle in a multi-ethnic population: A qualitative studyPLOS ONE

Dear Dr. Subramaniam,

Thank you for submitting your manuscript to PLOS ONE. After careful consideration, we feel that it has merit but does not fully meet PLOS ONE’s publication criteria as it currently stands. Therefore, we invite you to submit a revised version of the manuscript that addresses the points raised during the review process.

ACADEMIC EDITOR:The manuscript was improved, but there are still some issues that need to be properly addressed. Please see the comments in the attached file. Make sure you reply to all comments, including those left in the previous response letter to reviewers.

We look forward to receiving your revised manuscript.

Kind regards,

Eliana Carraça

Academic Editor

PLOS ONE

---

## [Author Response · Author response to Decision Letter 1]

18 Oct 2022

15 October 2022

Eliana Carraça

Academic Editor

PLOS ONE

Ref: PONE-D-22-13181R1

Barriers and Facilitators for adopting a healthy lifestyle in a multi-ethnic population: A qualitative study

Dear Dr. Carraca

We would like to thank you and the reviewers for the second review. We have addressed the points raised by the reviewers in the revised manuscript as tracked changes. Our replies to their comments are attached and highlighted in bold for easy reference in our reply.

Academic Editor

The manuscript was improved, but there are still some issues that need to be properly addressed. Please see the comments in the attached file. Make sure you reply to all comments, including those left in the previous response letter to reviewers.

We have addressed all the comments in the attached file.

Reviewer: 1

(Comments in pdf file and response to our Reply to the reviewers)

We want to thank the reviewer for once again going through our article carefully and providing us with invaluable insights. The changes suggested in the language have been incorporated directly as tracked changes in the manuscript.

We have analysed the themes by socio-demographic groups and presented the data.

We hope we have addressed the reviewers’ comments adequately and look forward to a favourable reply.

Regards

Authors

---

## [Editor Report · Decision Letter 2]

20 Oct 2022

Barriers and Facilitators for adopting a healthy lifestyle in a multi-ethnic population: A qualitative study

PONE-D-22-13181R2

Dear Dr. Subramaniam,

We’re pleased to inform you that your manuscript has been judged scientifically suitable for publication and will be formally accepted for publication once it meets all outstanding technical requirements.

Kind regards,

Eliana Carraça

Academic Editor

PLOS ONE
---

## [Editor Report · Acceptance letter]

24 Oct 2022

PONE-D-22-13181R2 

Barriers and Facilitators for adopting a healthy lifestyle in a multi-ethnic population: A qualitative study 

Dear Dr. Subramaniam:

I'm pleased to inform you that your manuscript has been deemed suitable for publication in PLOS ONE. Congratulations! Your manuscript is now with our production department. 

Kind regards, 

on behalf of

Dr. Eliana Carraça 

Academic Editor

PLOS ONE